# Research on Manufacturing Equipment Operation State Evaluation Technology Based on Fractional Calculus

**DOI:** 10.3390/s23073373

**Published:** 2023-03-23

**Authors:** Yanhong Zuo, Hua Cheng, Guoqing Geng, Shilong Xia, Chao Zhou

**Affiliations:** 1School of Mechanical and Electrical Engineering, Anhui Jianzhu University, Hefei 230601, China; zuoyh626@sohu.com (Y.Z.);; 2School of Electrical Engineering and Automation, Anhui University, Hefei 230601, China

**Keywords:** manufacturing equipment, fractional order calculus, effective evaluation, operational status

## Abstract

The operational status of manufacturing equipment is directly related to the reliability of the operation of manufacturing equipment and the continuity of operation of the production system. Based on the analysis of the operation status of manufacturing equipment and its characteristics, it is proposed that the concept of assessing the operation status of manufacturing equipment can be realized by applying the real-time acquisition of accurate inspection data of important parts of weak-motion units and comparing them with their motion status evaluation criteria. A differential data fusion model based on the fractional-order differential operator is established through the study of the application characteristics of fractional-order calculus theory. The advantages of Internet of Things (IoT) technology and a fractional order differential fusion algorithm are integrated to obtain real-time high-precision data of the operating parameters of manufacturing equipment, and the research objective of the operating condition assessment of manufacturing equipment is realized. The feasibility and effectiveness of the method are verified by applying the method to the machining center operation status assessment.

## 1. Introduction

Mechanical manufacturing equipment is an indispensable tool in industrial production, and the reliability of manufacturing equipment is a prerequisite for the normal operation of an enterprise’s manufacturing system. Therefore, realizing a real-time and accurate assessment of the operational status of manufacturing equipment has become an urgent problem in enterprise production. If we can achieve real-time accurate assessment of the equipment’s operating status, we can scientifically develop equipment maintenance programs and equipment failure points in advance to predict and prevent issues. To ensure the reliability of the enterprise manufacturing equipment management system and the normal operation of enterprise production, and promote the development of the manufacturing industry, is of great significance.

According to the operation rule and lifecycle of manufacturing equipment, the operation of manufacturing equipment can be divided into three stages: normal operation, fault warning, and fault state. The evaluation of the equipment operation state at home and abroad primarily focuses on the diagnosis stage of a fault. For example, Maliuk et al. proposed a Gaussian mixture model-based (GMM) bearing fault band selection (GMM-WBBS) method for signal processing, which benefits reliable feature extraction using a fault frequency-oriented Gaussian mixture model (GMM) window series [1]. Stator winding faults, broken rotor bar faults, rotor asymmetry, and abnormal air-gap eccentricity have been used in the diagnosis of bearing faults [2,3,4]. Krzysztof Kecik et al. [5] presented the problem of rolling bearing fault diagnosis based on a vibration velocity signal. To determine an optimal set of fault-relevant variables, Deng et al. [6] proposed a fault reconstruction algorithm based on least angle regression (LARS). Referance [7,8] used a contribution plot algorithm to identify the fault-relevant variable without any prior information about the fault. Wang et al. applied the ADD-HMM algorithm to predict the occurrence of mechanical equipment faults [9]. Reng et al. implemented a deep-learning waveform image recognition method for the fault diagnosis technology of bearings [10]. Yong et al. applied a Bayesian network to the fault diagnosis of weapons and equipment [11]. Bin et al. applied manic power variability to determine the failure time of mechanical equipment [12]. Qi et al. [13] proposed a progressive fault diagnosis method (PFDM) for the overall diagnosis of an entire EHA system, which significantly improved the safety and reliability of the double-redundancy EHA system. Zhou et al. [14] applied multiple states to evaluate the operation status of manufacturing systems; however, the method was based on the ideal state and failed to consider the complexity and variability of the manufacturing equipment use environment.

It can be seen from the existing research that the proposed methods have the following characteristics.

Limitations of application objects: Each fault diagnosis method mentioned above can only be applied to specific research objects, and cannot be extended to the diagnosis of various faults in other manufacturing equipment.Limitations of application functions: In equipment management, there are no real-time online monitoring and fault prediction functions that affect the reliability of equipment operation and the continuity of production system work.

Therefore, there is an urgent need for a method that can monitor and evaluate the running status of various online equipment in the current manufacturing systems. This method should have the function of equipment operation state judgment and fault prediction in advance and ensure the normal operation of equipment and production systems through the timely troubleshooting of equipment faults. This study draws on the research results of various experts and scholars in the evaluation of the operational status of different types of equipment. To analyze the working characteristics of various types of manufacturing equipment, we explore evaluation methods that can be adapted to the operating status of multiple types of manufacturing equipment, combine the advantages of IoT technology and data processing technology, and achieve the research goal of real-time and accurate evaluation of the operating status of various types of manufacturing equipment.

## 2. Manufacturing Equipment Operating Condition Assessment Methods

### 2.1. Fundamental Principle

Manufacturing equipment is a combination of multiple motion units, each of which has its own motion law and parameters. Many moving segments exist as long as there is a moving segment in a state of failure, and the equipment fails. Therefore, the operational status of manufacturing equipment depends on the shortest life-cycle of all motion units. As part of the manufacturing equipment, the working state of the motion unit is divided into three stages: normal operation, fault warning, and fault state. The motion parameters under each working state have their own value intervals: [0, *C*_1_], [*C*_1_*, C*_2_], and [*C*_2_*, C*_3_]. If we can obtain accurate operating parameters *C* of the motion unit in real time, we can accurately judge the operating state of the motion unit or even the entire equipment in real time by referring to the value intervals of each working state.

### 2.2. Implementation Steps of the Method

According to the evaluation principle of the operational state of the manufacturing equipment, the evaluation of the operational state of the entire equipment can be divided into the following steps:The main motion units that affect the running state of the equipment were analyzed according to the type of manufacturing equipment.The operation law of the main motion units was analyzed, and the motion unit with the shortest life-cycle was selected.The main parameters affecting the running state of the motion unit and their value ranges under different working states were analyzed.High-precision detection data for important motion parameters can be obtained when the motion unit operates normally in real time.The running state of the motion unit was determined by comparing the high-precision parameter values obtained in real time with the above value range.

### 2.3. Premise of Method Implementation

According to the introduction of the basic principle and implementation scheme of the evaluation method of manufacturing equipment operation status, it can be seen that the implementation of the scheme needs to solve the following three problems.

The judgment technology of the shortest life-cycle motion unit among many motion units;The real-time detection technology for the main parameters affecting the running state of the motion unit;The high-precision acquisition technology of the main detection information data in a complex environment.

Among the above three preconditions, the judgment technology of the shortest life-cycle motion unit in the motion unit can realize the judgment function of the minimum life-cycle motion unit of different manufacturing equipment by analyzing the motion characteristics of the motion unit and the accumulation of equipment management experience. In the era of rapid development of information network technology, there are no technical problems when using IoT to obtain all types of detection information and data in real time; however, in the collection of detection information data, the detection value of the information will inevitably be affected by factors such as equipment performance, the working environment, and signal interference, which will cause detection errors in information data, resulting in the acquisition technology of high-precision information data becoming the main problem in the implementation of manufacturing equipment operation status evaluation schemes.

## 3. High-Precision Information Data Acquisition Technology

The essence of the hardware method is to improve the detection accuracy of the information data using high-performance detection instruments. The main research results for the hardware methods are as follows: Hu et al. [15] proposed a high-precision safety valve test architecture with three testing channels and effectively solved the problems of current safety valve testing. Huijun et al. [16] developed a comprehensive sliding-separation test platform for RV reducers to realize the high-precision and high-display test performance for various RV reducer parameters. References [17,18] proposed a load differential radiation pulse on a transient electromagnetic high-performance radiation source for pulse-scanning detection to solve the problems of urban electromagnetic interference and insufficient harmonic components emitted by radiation sources. References [19,20,21] designed a hardware system based on radar and realized a real-time detection function for underground space-related information by enlarging the detection information. Jiaqi et al. [22] proposed a one-stage remote sensing image object detection model: a multi-feature information complementary detector (MFICDet), which can improve the ability of the model to recognize long-distance dependent information and establish spatial–location relationships between features.

However, in engineering applications, we found that the hardware method had the following shortcomings.

The detection accuracy of information data depends on the performance of the detection equipment. With improvements in detection accuracy, the cost of the detection system is higher. Therefore, they exhibit low-cost performance.Their essence is to reduce the signal distortion caused by energy loss and signal interference in the information transmission process by improving signal strength. However, when collecting information, the measurement error of the information data cannot be eliminated owing to the differences in the equipment performance and working environment.

In recent years, most researchers have attempted to use software methods to achieve high-precision information data detection to solve the shortage of hardware methods for information data detection in complex environments. The essence of the software method is the information data-fusion algorithm. To date, many studies have been conducted on this topic. Common mathematical algorithms are fuzzy set theory [23], fuzzy neural networks [24], the probability model [25], and the particle swarm optimization algorithm [26]. For example, Huo et al. [27] proposed an integral infinite log-ratio algorithm (IILRA) and an integral infinity log-ratio algorithm based on signal-to-noise ratio (BSNR-IILRA) to improve the detection accuracy of the laser communication detection position in the atmosphere. Zhiyuan et al. [28] proposed a normalized-variance detection method based on compression sensing measurements of received signals and solved the problem of fast and accurate spectrum sensing technology under the condition of a low signal-to-noise ratio. Liu et al. [29] proposed a target detection algorithm based on improved RetinaNet, which is suitable for transmission-line defect detection and improves the intelligent detection accuracy of UAV in power systems. Cheng et al. [30] proposed a lightweight ECA-YOLOX-Tiny model by embedding an efficient channel attention (ECA) module into it, which has a higher response rate for decision areas and special backgrounds, such as overlapping small target insulators, insulators obscured by tower poles, or insulators with high-similarity backgrounds. Liu Wenqiang et al. [31] introduced a point cloud segmentation and recognition method based on three-dimensional convolutional neural networks (3-D CNNs) to determine the different components of the catenary cantilever devices. Yin et al. [32] proposed a complementary symmetric geometry-free (CSGF) method that made the detection of cycle slips more comprehensive and accurate. Lingfeng et al. [33] established a junction temperature model based on a multiple linear stepwise regression algorithm, and used it to extract high-precision intersection online temperatures. However, through the analysis of various current software methods, the following deficiencies were found in the detection of information in complex environments:They do not improve the strength of the detection information and cannot solve the problems of energy loss and signal interference during information transmission. Therefore, it is difficult to apply these methods to engineering practice.These methods do not analyze the cause of the information data detection error, the change rule of each influencing factor, or its influence on the detection value. Therefore, it is difficult to improve the detection accuracy of information data by reducing the detection error caused by various influencing factors.

That being the case, we conclude that an ideal high-precision detection method for information data under the joint action of multiple influencing factors has not yet been developed. To solve these problems, our team has been using the method of fractional calculus theory in data processing for many years [34,35,36,37,38,39] and found that fractional differential operators are suitable for studying nonlinear, non-causal, and non-stationary signals, and have the dual functions of improving detection information and enhancing signal strength. Therefore, by fusing the differences between the information and data, the information and data detection errors caused by various influencing factors can be eliminated. By improving the signal strength of the information, it can compensate for the energy loss of the signal in the transmission process and improve the anti-interference ability of the signal. Therefore, in this paper, we try to combine fractional order calculus with IoT technology to realize the function of real-time accurate evaluation of the operation status of a manufacturing assembly.

## 4. Fractional Order Differentiation

### 4.1. Definition of Fractional Order Differentiation

Fractional differentials, also known as fractional derivatives, extend the differential order of integer-order differential equations to a fractional order. Fractional differentials (FDs) emerged in 1812. In hundreds of years of development, many scholars have proposed their own definition methods and theoretical systems based on their own understanding and application fields. Therefore, a strict definition of the fractional differential is not yet available. Currently, the most commonly used definitions are those of Grunwald–Letnikov (G–L), Caputo, and Riemann–Liouville (R–L). Assuming that the information acquisition system collects any energy signal *f*(*t*) and *f*(*t*) ∈ L^2^ (R), we can obtain the three kinds of fractional differentials of signal *f*(*t*) as follows:

#### 4.1.1. R–L Definition of the Fractional Differential of the Fractional

According to the principle of mutual inverse operation of Cauchy’s indefinite integral formula with fractional differentiation and fractional integration, we can obtain the R–L definition of fractional differentiation:(1)Datvft=dndtnDav−nft=1Γn−vdndtn∫att−τn−v−1fτdτ

In the equation above, 0 ≤ *n* − 1 < *v* < *n*, *τ* is the signal frequency, and *τ* ∈ [*a*, *t*], Γ is the gamma function.

The main advantage of this definition is that the initial value of the Laplace transform can be obtained using only an integer-order derivative. Its disadvantage is that it has stricter requirements for function *f*(*x*) than other definitions, but its premise is that the integer-order derivative of function *f*(*x*) is absolutely integrable.

#### 4.1.2. G–L Definition of the Fractional Differential of the Fractional

For any real number *v*, where the integer part of *v* is denoted as [*v*], assuming that the function $f\left(t\right)$ has (*n* + 1)-order continuous derivatives within the interval [*a*, *t*], when *v* > 0 and *n* ≥ [*v*], the fractional *v*-order derivative is defined as follows:(2)Datvft=limh→0fhvt=limh→0h−v∑j=0t−ah−1jvjft−jh

In the above equation, 0 ≤ *n* − 1< *v* < *n*, *h* is the value of the step, and $\left(\begin{array}{c} v\\ j \end{array}\right)$ is a binomial coefficient, which is given by
(3)$$\left(\begin{array}{c} v\\ j \end{array}\right)=\frac{v\left(v-1\right)\left(v-2\right)\ldots \left(v-j+1\right)}{j!}=\left[\begin{array}{c} -v\\ j \end{array}\right]$$

Based on the classical definition of the integer derivative of a continuous function, this definition extends the order of the differential from integer to fraction, which is suitable for numerical calculations.

#### 4.1.3. Caputo Definition of the Fractional Differential of the Fractional

The equation is as follows:(4)DaCtvft=Dat−n−vDnft

In the equation above, 0 ≤ *n* − 1 < *v* < *n*.

To simplify the calculation of the fractional differential, this definition further improves the Grunwald–Letnikov definition based on the basic properties of fractional calculus.

Among the three definitions, the G–L definition is widely used in engineering because of its simple calculation process and high speed. Therefore, the G–L definition of fractional partial differentials was used in this study to research manufacturing equipment evaluation technology.

### 4.2. Properties and Applications of Fractional Order Differentiation

Detection data were obtained from the analysis and processing of the detection information. Therefore, the measurement error generated during the acquisition and transmission of detection information directly affects the accuracy of detection data. In terms of the mathematical properties of the signal and the characteristics of the signal structure, the detection signal contains information on the fractional differential characteristics; however, this type of information is not suitable for processing by integer differential operators. For years, many mathematicians and scientists have attempted to apply fractional calculus theory in signal processing and have achieved good application results. It is assumed that there is a detection signal $S\left(t\right)$, where $S\left(t\right)\in L^{2}\left(R\right)$ and its Fourier transform is
(5)$$S\left(\hat{\omega }\right)=\int_{R}S\left(t\right)e^{-i\omega t}dt$$

Let $S^{v}\left(t\right)$ be the *v*-order differential of $S\left(t\right)$. According to the properties of the Fourier transform, we know that the *v*-order differential operator is equal to the multiplicative operator of the *v*-order differential multiplier function, $\hat{d}(\omega )={\left(i\omega \right)}^{v}$. Thus, we obtain the following equation:(6)$$D^{v}S\left(t\right)\overset{FT}{\underset{}{\Leftrightarrow }}{\left(\hat{D}S\right)}^{v}\left(\omega \right)={\left(i\omega \right)}^{v}S\left(\hat{\omega }\right)={\left|\omega \right|}^{v}e^{i\theta^{v}\left(\omega \right)}S\left(\hat{\omega }\right)$$

From the perspective of signal modulation, the physical meaning of the fractional differential of a detection signal is equivalent to the generalized amplitude and phase modulation. From the perspective of signal processing, the *v*-order fractional calculus operation of the detected signal is equivalent to establishing a linear time-invariant filtering system for the signal, and its filtering function is:(7)$$d^{v}(\hat{\omega })={\left[\omega \right]}^{v}e^{i\theta_{v}\left(\omega \right)}$$

Through the Fourier transform defined by the G–L of fractional-order calculus, it is known that the essence of fractional-order calculus processing of the energy signal is to filter the signal, and its filter function is $j^{v}\left(\hat{\omega }\right)={\left(i\omega \right)}^{v}={\left|\omega \right|}^{v}e^{i\theta_{v}\left(\omega \right)}$. According to the filter function, we can draw the spectral characteristic curves of the fractional-order differential operator and fractional-order integral operator during signal processing, as shown in Figure 1. After analyzing the spectral characteristic curves, we obtained the following characteristics of fractional-order calculus in the signal processing.

From Equation (7), we know that, after fractional differential processing, the signal has the following characteristics:The signal shows different levels of signal enhancement for different fractional differential operators, so that the very-low-frequency components of the signal can be preserved nonlinearly.From a physical perspective, signal processing by the fractional differential operator can be understood as the generalized amplitude phase modulation of the signal. Thus, the fractional differential operator can significantly improve signal strength when processing the high-frequency part of the signal.The fractional differential operator significantly improved the high-frequency signal strength. It also improved low-frequency signals.

## 5. Fractional Order Differentiation Based Fusion Model of Equipment Operation Parameters

### 5.1. Characteristics of Manufacturing Equipment Inspection Information Data

Currently, manufacturing systems are divided into two modes: continuous and discrete manufacturing. Continuous manufacturing is mainly applied to the production modes of small and large batches. During the production process, the operational status of the motion unit changes regularly. As long as the change rule of the operation status and the evaluation criteria of each operation status are mastered, the evaluation function of the operation status of the continuous manufacturing equipment can be realized. Discrete manufacturing is a multi-variety and small-batch production mode, and the difference in product types leads to differences in the operating status of manufacturing equipment during the manufacturing process. Therefore, the operation status evaluation scheme of continuous manufacturing equipment cannot be applied to the evaluation of the operation status of discrete manufacturing equipment. Aiming at the important problem that the operation status of manufacturing equipment in the two types of manufacturing is quite different, and the traditional evaluation technology is difficult to consider, this study explores a method to realize the evaluation function of the operation status of manufacturing equipment in the two types of manufacturing by comparing the real-time acquisition of accurate motion parameters and the evaluation criteria without considering the motion rules of manufacturing equipment.

### 5.2. Fractional Order Differential Based Operational State Evaluation Model

According to the spectral characteristics of fractional calculus in information processing, it is known that both fractional calculus and integral operators have the advantage of reducing the measurement error caused by external interference signals, and both can effectively fuse to eliminate the information data detection error caused by a single influencing factor in a one-dimensional space. However, the fractional-order differential algorithm has the advantage of enhancing the signal strength and reliability of the detection system compared to the fractional-order integral algorithm. Therefore, the fractional-order differential algorithm is suitable for processing the main parameter values of the motion unit of manufacturing equipment.

After analyzing the common fault types of manufacturing equipment, we can obtain the main fault types and their main causes, analyze the motion unit where the fault point is located, and obtain the main influencing factor *x* of its operation status. Then, the value interval [*a*, *b*] of the influencing factor *x* and its corresponding detection value *S_i_* in the interval after the experiment fit the function formula *S*(*x*) between the influencing factor *x* and the detection value *S*. The mathematical model of the manufacturing equipment operation status evaluation is shown in Equation (8) according to Equation (2).
(8)DaxvSx=limh→0Shvx=limh→0h−v∑i=0b−ah−1iviSx−ih≈Sx−vSx−h−v1−v2!Sx−2h+⋯+v1−v(2−v)⋯(n−1−v)n!Sx−nh

## 6. Application of Assessment Methods

### 6.1. Experimental Platform Construction

According to the analysis of the types and causes of manufacturing equipment failures over the years, it can be seen that the common failure of manufacturing equipment is the decline in the movement accuracy of the spindle components, which leads to the equipment’s failure to meet the processing accuracy requirements. After analyzing the causes of various machine tool spindle unit failures, it was concluded that the main reason is that the fatigue damage of the spindle bearing brings a bearing clearance greater than the maximum allowable value, resulting in a machine tool machining accuracy below the machining accuracy standards. Most bearings used in machine tool spindle components are rolling bearings, which have the performance characteristics of easy wear and a short life-cycle in the application, thus becoming the main source of spindle motion unit failure. The causes of the short bearing life-cycle are fatigue damage and permanent deformation; however, fatigue damage is the main cause of bearing failure, which is the result of its accumulation owing to the amount of wear during normal operation. Therefore, only by obtaining the actual wear amount of spindle bearings in real time and referring to the value interval of each operating condition can the evaluation function of the operating condition of the manufacturing equipment be obtained.

The bearings used in the spindle motion unit of the equipment were the NN3046 series cylindrical roller bearings. Because the equipment is used in discrete manufacturing systems, the wear of bearings presents the characteristics of irregular and dynamic changes, which makes the traditional method of applying the change law of motion parameters to evaluate the equipment operation state appear powerless. Therefore, this study attempts to verify the feasibility and reliability of the application of this method in the evaluation of the equipment operation state.

### 6.2. Information Data Collection

#### 6.2.1. Data Collection Methods

The best solution for measuring the parameter value of the object is to place the corresponding sensor at the corresponding position of the object; however, in the detection of rolling bearing wear of precision machine tools, the existence of disassembly of the main moving parts of the machine tool leads to a decrease in accuracy. The detection value of the clearance between the outer ring of the rolling bearing and the spindle is affected by roller interference and other factors, making it difficult to implement the above method in the detection of spindle bearing wear of precision machine tools. Because the front bearing of the machining center is the component with the greatest force and wear in the main motion system, based on the analysis of the structure of the main drive system, the experiment used a displacement sensor arranged on the inside of the front bearing end cover of the spindle near the spindle, and indirectly realized the detection function of the front bearing wear by detecting the value of the external runout of the spindle during operation. Because it is difficult to predict the magnitude and direction of the force on the spindle during machining and the speed is too fast to result in the loss of experimental data owing to the lack of sensitivity of the testing instrument, the experiment was carried out under idling conditions with a spindle speed of 750 r/min. When the spindle is idle, the clearance caused by bearing wear is reflected in the upper part of the bearing bore under the action of self-weight. Therefore, the effective detection of bearing wear can be realized by arranging displacement sensors only in the upper part of the spindle. Based on a comprehensive consideration of the installation space and detection accuracy, five L3002-12.7 LVDT displacement sensors were placed in the upper part of the bearing-end cover, the configuration of which is shown in Figure 2.

To ensure real-time detection data, the experiment used IoT technology to transmit the information collected by each sensor in real time. IoT technology has two methods for the transmission of detection information: wireless and wired networks. Wireless networks have the advantage of real-time collection of mobile equipment production information, but also have a shortage of information detection errors, which are suitable for the real time collection of production information of fixed equipment in close proximity, and have the double advantage of real time collection and accuracy in the collection of fixed equipment production information. Combining the advantages and disadvantages of the two transmission methods and the working characteristics of the equipment in this case, a wired network was applied to transmit the real time collected bearing clearance information to the information processing center for the expert system to analyze and judge.

#### 6.2.2. Experimental Data Collection

In a limited area, the variability of similar testing information data mainly originates from variability in the performance of the testing equipment. Although the standard deviation is an important indicator reflecting the performance of the testing equipment, the performance of each sensor changes dynamically owing to the influence of the service life and working environment. To obtain the true standard deviation of the sensors during the test, the equipment spindle at 750 r/min idle speed, the application of each sensor in the C-point position to collect test data, the spindle every two revolutions sampled a total of six times, and the relevant test data and its standard deviation values are listed in Table 1.

After obtaining the standard deviation of each sensor, the spindle continues to rotate, and the five sensors are sampled six times at the detection points A, B, C, D and E according to the above sampling frequency, and the average of the six detection values is taken as the detection value of the sensors at each point. The detection values of each sensor at different points are shown in Table 2.

### 6.3. Analysis and Processing of Detection Data

Although the data shown in Table 2 can be intuitively seen, sampling point D is the location of the largest amount of wear, because the differences in the performance of the testing equipment lead to a large difference between the detection data values, resulting in a large error between the detection data of different sensors at the same sampling point, such as sampling point A, where there are 2# and 4# sensor measurement errors of more than 5%; therefore, the data in Table 1 are not easy to use in the evaluation of the wear amount. That being the case, the evaluation of spindle bearing wear must be based on the high accuracy of the measured values at each sampling point to lay the foundation for the comparative method in the identification of the maximum wear point and to avoid misjudgment in other cases.

#### 6.3.1. Selection of the Influence Factor of the Detection Value

According to the above analysis, information detection data in the collection process will be affected by a variety of factors, such as equipment performance, the working environment, and signal interference; however, whether it is mobile equipment or fixed equipment, its working interval is within a limited range, and the working environment and signal interference factors on the same detection value are basically the same. Therefore, the variability between the detection data mainly comes from the variability between the performance of the detection equipment. Considering that the standard deviation is the best parameter for measuring the performance of the testing equipment, the standard deviation *S_i_* of the sensor can be used as the influencing factor *x* of the bearing clearance detection value. The influence of the influencing factor on the detection value of each testing point can be explored according to the standard deviation of each sensor and its measured value at each testing point, as listed in Table 2.

#### 6.3.2. The Functional Relationship between the Detection Value *F_i_* and the Impact Factor *x_i_*

Among various fitting algorithms, the least-squares method has the advantage of not requiring a priori data information in the data processing process. It is widely used to fit the function polynomial of the measurement data and can obtain the ideal data fusion accuracy, which is suitable for fitting the equation of the function *F*(*x*) between the detected value *F_i_* and each influence factor *x_i_* (standard deviation *S_i_*) at each point in the experiment; assuming that the expression of the function *F*(*x*) is
(9)$$F(x)=a_{0}+a_{1}x+a_{2}x^{2}+\cdots +a_{n}x^{m}$$

According to Equation (9), we can determine the parameters in the equation to fit the required functional relationship: According to the data values shown in Table 2, the polyfit function in MATLAB software can be applied to fit the values of each sampling point, as shown in Equation (10), and the functional relationship between each influence factor *x_i_ F_a_*(*x*), *F_b_*(*x*), *F_c_*(*x*), *F_d_*(*x*), and *F_e_*(*x*).
(10)$$\left\{\begin{array}{l} F_{a}(x)=-3.6416x+0.1269\\ F_{b}(x)=1.0405x+0.1562\\ F_{c}(x)=-3.0058x+0.1874\\ F_{d}(x)=1.0116x+0.2027\\ F_{e}(x)=5.0x+0.1265 \end{array}\right.$$

#### 6.3.3. Selection of Fractional Order *v* and Step Size *h* Values

As shown by Equation (8), establishing a detection data fusion model based on fractional differentiation under the definition of G–L requires two problems to be solved: determination of the fractional order *v* and the selection of step *h*.

Selection of order *v*:

After analyzing the amplitude-frequency characteristics of the fractional-order differential operator, it can be seen that when the differential order *v* ∈ [0,1], the signal intensity in the high-frequency stage increases with an increase in the fractional order. However, with an increase in the fractional order, the difference between the differential operators of different orders in the signal enhancement value shows a decreasing trend. Therefore, from the viewpoint of saving space and facilitating calculations, this experiment explores the application effect of the fractional-order differential operator in the processing of machining center spindle bearing wear detection data when the fractional order *v* is taken as 0.5, which is the middle value of [0,1].

2.Selection of step *h*:

During the processing of information data, the signal frequency depends on the value interval [a, b] of the influencing factor and its step value *h*. According to the amplitude-frequency characteristics of the fractional-order differential operator, it is known that the step length and value interval of the influence factor are associated with the frequency of the signal; however, in the high-frequency region of the signal, the difference between the signal enhancement effect of the differential operator in the same order is negligible. The smaller the step size *h*, the higher the fusion accuracy of the data; however, it also causes a decrease in the fusion efficiency owing to the increase in the computation step *n*. Therefore, the fusion speed is considered in this case. Considering both the fusion speed and fusion accuracy, the fusion step *h* = 0.0001 is taken according to the value range [0.0037,0.0048] of the influence factor *x* (sensor standard deviation) shown in Table 2, and the number of steps *n* = 11 must be calculated.

#### 6.3.4. Information Data Processing Techniques Based on Fractional Order Differential Operators

According to the fusion model of manufacturing equipment operation parameters under the G–L definition shown in Equation (3), the fusion processing model of the machining center spindle bearing wear data can be expressed as
(11)DaxvFkx=limh→0h−v∑i=0b−ah−1iviFkx−ih≈Fkx−vFkx−h−v1−v2!Fkx−2h+⋯+v1−v(2−v)⋯(n−1−v)n!Fkx−nh
where: *v* = 0.5, *n* = 11, *h* = 0.0001, *k* = a, b, c, d, e.

In Equation (11), *x* is the standard deviation $S_{i}$ of each sensor and $F_{k}$ is its detection value at each position. Now, the parameters (*v, h, n*) are substituted into Equation (6), and combined with Equation (5), and the detection data $F_{ij}$ of each sensor at different positions shown in Table 3 can be obtained using MATLAB (R2022a).

From the data shown in Table 3, it can be seen that the fused data values show two characteristics compared to the data before fusion processing: the measured values of different sensors at each detection point are evenly distributed around the mean value, the variability between the data is significantly reduced, the fused values increase significantly, and the amplification factor *K* is as high as 17.05 compared to that before fusion. It was verified that the fractional-order differential operator has amplitude-frequency characteristics that enhance the information’s intensity and reduce the variability among the information data. To compare the accuracy of the data before and after the fusion of the machining center spindle bearing wear, the data shown in Table 3 were divided by the amplification factor *K* to obtain the final value of the data shown in Table 2 after the fusion of the 0.5th order differential operator. The results are shown in Table 4.

### 6.4. Analysis of Operating Condition Assessment Results

#### 6.4.1. Evaluation Criteria for Bearing Wear

Machinery manufacturing equipment in the important parts of the life-cycle is equipped with normal operation [0,*T*_1_], failure warning [*T*_1_*,T*_2_], and failure state [*T*_2_*,T*_3_] in three time periods, corresponding to the main parameter values for [0,*C*_1_], [*C*_1_*,C*_2_], and [*C*_2_*,C*_3_]. The state for the standard parts and important parts of the main parameters have been developed to correspond to the value of the standard (parameter value *C ≤ C*_3_). However, the *C*_1_ and *C*_2_ parameter values have not been issued by the state corresponding to the value of the standard; each enterprise can only be determined according to the performance of the equipment and production characteristics. In this case, considering that the performance of the machining center is slightly higher than that of similar machine tools, and it is mainly used for processing small- and low-precision parts, with reference to the national standard of *C*_3_ ≤ 0.3 mm for the bearing clearance of a diameter more than 100 mm, the parameters *C*_1_ = 0.280 mm and *C*_2_ = 0.295 mm were set in the experiment to judge the operation of the machining center under the existing conditions.

#### 6.4.2. Evaluation of Operation Status

Combining the data shown in Table 2 and Table 4, it can be seen that the standard deviation between the fused data is significantly reduced compared with that before fusion. As shown at Detection Point D, the standard deviation of the fused detection data is 0.0013, and the precision is close to five times that before fusion, which is significantly lower than 0.0055 of the least squares method and 0.0036 of the particle swarm optimization [26]. The data distribution diagram of the detection point D before and after fusion is shown in Figure 3. It can be observed that the data are randomly distributed near their average value after being processed by the 0.5-order differential operator, and the discreteness is significantly reduced, greatly improving the accuracy of the detection data of the spindle bearing wear of the machining center. According to the data shown in Table 4, the test data at test point D were significantly higher than the measured values at the other test points. According to the evaluation method of the fault points mentioned above, test point D is the maximum point of the wear amount of the spindle bearing of the machining center in this case, and the wear amount is δ_D_ = 0.215 mm. With reference to the above set operational state evaluation standard, the wear amount δ_D_ is within [0,0.28], indicating that the equipment is in normal operation.

### 6.5. Application Analysis of Experimental Results

Based on the preconditions for the implementation of the evaluation method of the manufacturing equipment operation state described above, this case, through the analysis of the operation characteristics of the current typical manufacturing equipment-machining center, we conclude that the wear value of the main shaft bearing is an important basis for evaluating the operation state of the machining center and for realizing the judgment function of the shortest life-cycle motion unit among many motion units. In view of the fact that the position of the equipment in this case is static during operation, the detection information transmission mode based on LAN was adopted, which avoids the lack of information distortion caused by various interference factors in the transmission process and realizes the real-time detection function of the manufacturing equipment movement unit parameters. Through research on fractional calculus theory, the application of the fractional differential operator has the multiple advantages of enhancing the signal strength, improving the accuracy of the detection data, and realizing the high-precision acquisition function of the detection information data. Therefore, the entire experimental process meets the three prerequisites for the evaluation of the manufacturing equipment’s operation status. Through the application of the method described in this paper, the research goal of the real-time and accurate evaluation of the manufacturing equipment’s operation status is achieved.

## 7. Conclusions

Through an analysis of the operation characteristics of the manufacturing equipment, it was concluded that the operation status of the manufacturing equipment depends on the operation characteristics of the weakest motion unit in the equipment. It is proposed that the application of real-time acquisition of accurate values of important motion parameters in the motion unit can realize the concept of real-time evaluation of the operational status of manufacturing equipment by comparing it with its evaluation criteria. Through the research of fractional calculus theory, this paper proposes a combination of IoT technology advantages and fractional differential algorithms; to use IoT technology to realize the real-time collection function of detection information data, to apply a fractional differential operator to significantly reduce the difference between the detection data, and to solve the technical problem of real-time acquisition of high-precision detection values for the main motion parameters of manufacturing equipment movement units, which lays a foundation for the implementation of real-time and accurate evaluation schemes for manufacturing equipment operation status. Through the application of a real-time evaluation experiment on a machining center’s running state, the effectiveness and feasibility of the method described in this paper, in engineering practice, was verified.

## Figures and Tables

**Figure 1 sensors-23-03373-f001:**
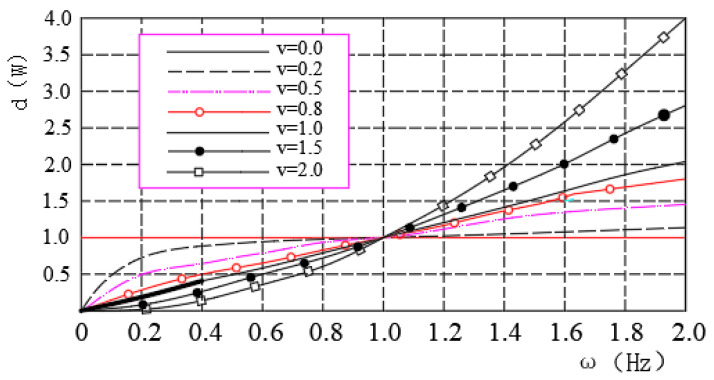
Amplitude-frequency characteristic curve of fractional-order differential operator.

**Figure 2 sensors-23-03373-f002:**
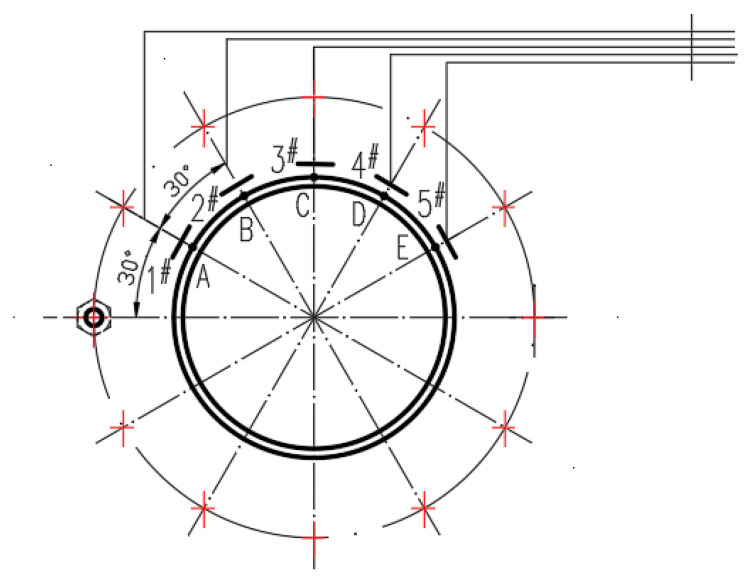
Diagram of the bearing wear testing equipment arrangement.

**Figure 3 sensors-23-03373-f003:**
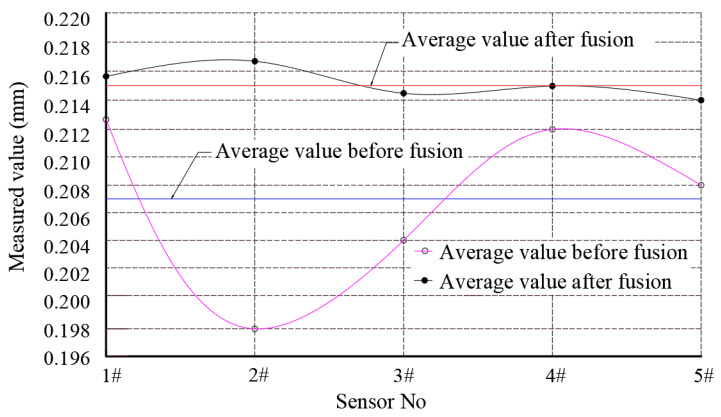
Distribution of D-point detection data before and after 0.5 order differential fusion.

**Table 1 sensors-23-03373-t001:** List of detection values and their standard deviations for each sensor at point C (mm).

Sensor No.	Number of Measurement						Mean Value *S_c_*	Standard Deviation *S_i_*
	1st	2nd	3rd	4th	5th	6th		
1#	0.164	0.158	0.168	0.166	0.171	0.158	0.1642	0.0048
2#	0.162	0.171	0.168	0.175	0.165	0.173	0.1690	0.0045
3#	0.162	0.173	0.165	0.163	0.171	0.165	0.1665	0.0041
4#	0.172	0.165	0.174	0.162	0.169	0.165	0.1678	0.0042
5#	0.168	0.162	0.165	0.163	0.165	0.173	0.1660	0.0037

**Table 2 sensors-23-03373-t002:** List of detection values of each sensor at different detection points and their average values (mm).

Location of Sampling Points	Sensor No.					$\mathrm{Average}\;\mathrm{Value}\;w_{i}$	Standard Deviation $\mathrm{before}\;\mathrm{Fusion}\;s_{wi}$
	1#	2#	3#	4#	5#		
A	0.112	0.105	0.108	0.118	0.114	0.1114	0.00454
B	0.158	0.162	0.165	0.162	0.156	0.1606	0.00320
C	0.175	0.172	0.178	0.171	0.177	0.1746	0.00273
D	0.213	0.198	0.204	0.212	0.208	0.2070	0.00632
E	0.152	0.147	0.145	0.149	0.146	0.1478	0.00248
Standard deviation of sensors $S_{i}$	0.0048	0.0045	0.0041	0.0042	0.0037	Total average $\bar{w}$	0.1603

**Table 3 sensors-23-03373-t003:** List of the fusion values of the detection values of each sensor at different detection points (mm).

Sampling Site Location	Sensor No.					$\mathrm{Average}\;\mathrm{Value}\;r_{j}$	Magnification Factor K
	1#	2#	3#	4#	5#		
A	1.7734	1.7918	1.8163	1.8102	1.8408	1.8065	17.05
B	2.7310	2.7258	2.7188	2.7205	2.7118	2.7216	
C	2.8543	2.8695	2.8897	2.8846	2.9099	2.8816	
D	3.6784	3.6939	3.6545	3.6694	3.6452	3.6683	
E	2.6244	2.5991	2.5655	2.5739	2.5318	2.5789	
Standard deviation of sensors $S_{i}$	0.0048	0.0045	0.0041	0.0042	0.0037	Average value after fusion $\bar{r}$	2.733

**Table 4 sensors-23-03373-t004:** List of the fusion results of the detection values of each sensor at different detection points (mm).

Sampling Site Location	Sensor No.	Average Value rj′	Standard Deviation after Fusion srj
1#	2#	3#	4#	5#
A	0.1040	0.1051	0.1065	0.1062	0.1080	0.1060	0.00144
B	0.1602	0.1599	0.1595	0.1596	0.1590	0.1596	0.00038
C	0.1674	0.1683	0.1695	0.1692	0.1707	0.1690	0.00111
D	0.2157	0.2178	0.2143	0.2152	0.2109	0.2148	0.00128
E	0.1539	0.1524	0.1505	0.1510	0.1485	0.1513	0.00184
Standard deviation of sensors Si	0.0048	0.0045	0.0041	0.0042	0.0037

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
