# Peer review of "Research on Manufacturing Equipment Operation State Evaluation Technology Based on Fractional Calculus"

_sensors, 2023, doi:10.3390/s23073373_

Round 1

Reviewer 1 Report

In this paper, in order to obtain real-time high-precision data, IoT technology and fractional order differential fusion algorithm are introduced. Some explanations should be given to the following comments:

1. The article mentions that the team has been using fractional calculus theory for many years, please explain the innovation of this article.

2. Please explain specifically that by fusing the differences between information and data, information and data detection errors caused by various influencing factors can be eliminated.

3. The article mentions the difference between continuous system and discrete system, and adopts the method of parameter comparison to determine the operating state, please explain how to obtain the operating state parameters for any device.

4. In order to improve the performance of the algorithm, please explain why the selection of v does not use the optimization method.

Author Response

Comment 1: In this study, the authors apply the fractional partial differential to develop a high-precision detection method for information data under the joint action of multiple influencing factors. However, the authors mainly described their conclusions and advantages. with previous studies but lacked the details of the application of fractional partial differential equations. The authors can supplement the derivation of the mathematical theories.

Response: The relevant content has been supplemented with references to reviewers' opinions. See chapter 3.1 on page 4 of the manuscript for details.

Comment 2:The article mentions that the team has been using fractional calculus theory for many years, please explain the innovation of this article.

Response: Based on previous research, we found that compared with other algorithms, the outstanding feature of the fractional differential algorithm is that it has the dual function of improving the information intensity and data accuracy. The purpose of this study was to effectively improve the anti-interference ability and energy loss of the detection signal while ensuring the accuracy of the information data and improving the accuracy of the equipment information data measurement.

Comment 3:Please explain specifically that by fusing the differences between information and data, information and data detection errors caused by various influencing factors can be eliminated.

Response: The essence of information data fusion is to use mathematical models to process information data, reduce the differences between information data affected by various factors, and achieve the purpose of improving the accuracy of information detection. The fusion effect depends on the progressiveness of the data fusion algorithm.

Comment 4:The article mentions the difference between continuous system and discrete system, and adopts the method of parameter comparison to determine the operating state, please explain how to obtain the operating state parameters for any device.

Response: Please forgive me for not explaining in detail in the article. Our original meaning is "to set the value range of parameters when the equipment is in different working states according to international standards or enterprise standards, and then obtain the working state of the equipment by comparing the detected parameter value with the standard value".

Comment 5:In order to improve the performance of the algorithm, please explain why the selection of v does not use the optimization method.

Response: As shown in Figure 1, the signal intensity in the high-frequency stage increases with an increase in the fractional order. However, with an increase in the fractional order, the difference between the differential operators of different orders in the signal enhancement value shows a decreasing trend. Therefore, from the viewpoint of saving space and facilitating calculations, the experiment explores the application effect of the fractional-order differential operator in the processing of machining center spindle bearing wear detection data when fractional order v is taken as 0.5. See red text on page 11 for relevant content.

Reviewer 2 Report

All the remakrs please find in the enclosed file

Author Response

Comment 1:Some editorial corrections should be made e.g.:

Response: We revised the content of the manuscript according to your suggestion. Please see the red text in the manuscript for further detail.

Comment 2:In the introduction part it is stated ”Limitations of application objects: Each fault diagnosis method mentioned above can only be applied to specific research objects, and cannot be extended to the diagnosis of various faults in other manufacturing equipment.”. In this context systemic approach to considered issue should be discussed.

Response: Thank you for this comment. From an objective point of view, we should use the limited literature to determine the current status of equipment fault diagnosis. Now, “Each fault diagnosis method can only be applied to specific research objects” is changed to “: Each fault diagnosis method mentioned above can only be applied to specific research objects”.

Comment 3:In chapter ”Premise of method inplementation” three main problem to solve were enumerated but one more problem exists. How to identify main information data in complex environment.

Response: Thank you for this comment. Signal recognition must be based on the characteristics of information. It is a complex scientific problem to judge by comparing the acquired signals. Because the research in this study is to detect the data and characteristics of a certain type of information, there is no need to identify the type of signal.

Comment 4:The symbols used in formulas should be explained (e.g. formula1)

Response: Thank you for your correction. The content on page 5 has been modified. Please check!

Comment 5:Chapter 5.1. and second paragraph of chaper 6.1. are not clear and should be rewritten.

Response: Thank you for your guidance. These contents have been modified according to your comments. Please check!

Reviewer 3 Report

1.In section 2.1(Pg.2), it is mentioned that “Many motion units exist as long as there is a unit in a state of failure, and the equipment will be in a state of failure.”, do you mean the manufacture equipment is a tandem system? 

2.In section 3(Pg.4), by fusing the differences between information and data, the information and data detection errors caused by various influencing factors can be eliminated. How can you make this conclusion?

3.In section 5.2(Pg.2), why is there a repetitive state that “Therefore, fractional-order differential or integral arithmetic can be applied to the processing of the main parameter values of the manufacturing equipment motion unit, and fractional-order differential arithmetic has the advantage of enhancing the signal strength and reliability of the detection system compared to fractional-order integral arithmetic, which has the disadvantage of weakening the information strength.”?

4.In section 6.1(Pg.8), it is mentioned that “According to the analysis of manufacturing equipment failure types and their causes over the years, the common failure of manufacturing equipment from the spindle motion unit cannot meet the processing accuracy requirements.”, on what analysis did you get this conclusion?

5.'Remaining useful life prediction and predictive maintenance strategies for multi-state manufacturing systems considering functional dependence' and other newly published related works from Sensors should be discussed.

Author Response

Comment 1:In section 2.1(Pg.2), it is mentioned that “Many motion units exist as long as there is a unit in a state of failure, and the equipment will be in a state of failure.”, do you mean the manufacture equipment is a tandem system?

Response: Thank you for this suggestion. The standard expression should be "sports link.” The error you have put forward has been modified. Please check the red text In section 2.1 (Pg. 2) for the modified content.

Comment 2:In section 3(Pg.4), by fusing the differences between information and data, the information and data detection errors caused by various influencing factors can be eliminated. How can you make this conclusion?

Response: Because the essence of information data fusion is to establish a mathematical model, reduce the impact of various factors on the detection information through the application of the model, reduce the differences between information data, and achieve the purpose of improving the accuracy of information detection. The fusion effect depends on the progressiveness of the data-fusion model.

Comment 3:In section 5.2(Pg.2), why is there a repetitive state that “Therefore, fractional-order differential or integral arithmetic can be applied to the processing of the main parameter values of the manufacturing equipment motion unit, and fractional-order differential arithmetic has the advantage of enhancing the signal strength and reliability of the detection system compared to fractional-order integral arithmetic, which has the disadvantage of weakening the information strength.”?

Response: Please forgive our negligence in the writing of the manuscript. We have revised this section according to your instructions. Please see the red text on page 7.

Comment 4:In section 6.1(Pg.8), it is mentioned that “According to the analysis of manufacturing equipment failure types and their causes over the years, the common failure of manufacturing equipment from the spindle motion unit cannot meet the processing accuracy requirements.”, on what analysis did you get this conclusion?

Response: Please forgive us with an inaccurate description in the manuscript. The question you have raised has now been revised. For the revised content, see the red text In section 6.1 (Pg. 8).

Comment 5:'Remaining useful life prediction and predictive maintenance strategies for multi-state manufacturing systems considering functional dependence' and other newly published related works from Sensors should be discussed.

Response: Thank you for pointing out the direction of future research. We have supplemented the analysis of the application characteristics of multiple states in the evaluation of the manufacturing system operation status. The literature [14] is also presented here.